# The Relationships between Caregivers’ Concern about Child Weight and Their Non-Responsive Feeding Practices: A Systematic Review and Meta-Analysis

**DOI:** 10.3390/nu14142885

**Published:** 2022-07-14

**Authors:** Jian Wang, Xiaoxue Wei, Yan-Shing Chang, Ayako Hiyoshi, Kirsty Winkley, Yang Cao

**Affiliations:** 1Florence Nightingale Faculty of Nursing, Midwifery and Palliative Care, King’s College London, London SE1 8WA, UK; yan-shing.chang@kcl.ac.uk (Y.-S.C.); kirsty.winkley@kcl.ac.uk (K.W.); 2School of Nursing, Shanghai Jiao Tong University, Shanghai 200025, China; weixiaoxue1998@163.com; 3Clinical Epidemiology and Biostatistics, School of Medical Sciences, Örebro University, 70182 Örebro, Sweden; ayako.hiyoshi@oru.se; 4Unit of Integrative Epidemiology, Institute of Environmental Medicine, Karolinska Institutet, 17177 Stockholm, Sweden

**Keywords:** children, caregivers, weight concern, feeding practices, systematic review, meta-analysis

## Abstract

Background: It is unclear whether caregivers’ concern about child weight impacts their non-responsive feeding practices. This systematic review aimed to examine their relationships. Methods: A systematic search of PubMed, Embase, PsycINFO, Web of Science core collection, CINAHL and grey literature was conducted from inception to March 2022, following PRISMA guidelines. Data synthesis was performed using a semi-quantitative approach and a meta-analysis. Results: A total of 35 studies with 22,933 respondents were included in the review for semi-quantitative analyses. Thirty-four studies examined 52 associations between concern about child weight and restriction with 40 statistically significant associations being observed. A total of 34 relationships between concern about child weight and pressure to eat were investigated, with 12 being statistically significant. The pooled regression coefficients (β) demonstrated that caregivers’ concern about child overweight was positively associated with restriction (β = 0.22; 95%CI: 0.12, 0.31), negatively associated with use of food as a reward (β = −0.06; 95%CI: −0.11, −0.01), and not statistically associated with pressure to eat (β = −0.05; 95%CI: −0.13, 0.04). The pooled odds ratios (ORs) indicated that caregivers who were concerned about child overweight were found to use restrictive feeding more often (OR = 2.34; 95%CI: 1.69, 3.23), while less frequently adopting pressure to eat (OR = 0.76; 95%CI: 0.59, 0.98) compared with those without concerns. The results also showed that caregivers who were concerned about child underweight were more likely to force their children to eat (OR = 1.83; 95%CI: 1.44, 2.33) than those without concerns. Conclusion: Caregivers’ concern about child weight may be an important risk factor for non-responsive feeding practices. Thus, interventions are needed to focus on managing and relieving caregivers’ excessive concern about child weight, especially overweight, which may optimize feeding practices and subsequently contribute to child health.

## 1. Introduction

The rise of overweight and obesity in children has become a significant public health issue, which affected 39 million children under the age of 5 and over 340 million children and adolescents aged 5–19 in 2020 worldwide [1]. Childhood overweight and obesity commonly lead to adulthood obesity and increase the risk of chronic diseases such as cardiovascular disease and type 2 diabetes [2,3,4].

Previous evidence has conceptualized contributors to childhood overweight and obesity through multiple levels (e.g., genetic, and environmental factors) [5,6,7]. Among these risk factors, caregivers’ feeding practices have been shown to play an important role [8,9,10,11]. There are two types of feeding practices: non-responsive and responsive feeding [12,13,14]. Non-responsive feeding practices (also known as coercive control), such as forcing to eat, restricting food, and using food as a reward [15], have been widely studied and raised crucial concerns owing to their close links with childhood obesity [8,9,10,11,16]. Positive relationships between non-responsive feeding practices and child weight status have been consistently reported [8,16]. A recent meta-analysis from 51 studies, with 17,431 parent-child dyads, reported that the use of controlling feeding practices by caregivers was associated with a greater risk of childhood obesity [8]. Caregivers’ feeding practices are therefore critical in addressing childhood overweight and obesity [17,18].

Costanzo and Woody (1985) proposed that parents were more likely to exert a higher level of external control over children’s eating (e.g., pressure to eat, food restriction) when they were concerned about their child’s weight [19,20]. Empirical evidence has also confirmed that caregivers’ concern about child weight plays a key role in whether a caregiver may use non-responsive feeding practices (i.e., coercive control). For instance, a cross-sectional study (*n* = 273) in the United States (US) indicated that parental higher restrictive feeding was associated with more concern about their preschool children’s weight [21]. Similarly, a population-based study (*n* = 1284) reported that maternal concern about their four- to seven-year-old children’s weight was positively associated with restrictive feeding and negatively associated with pressure to eat [22]. On the other hand, Costa et al. [23] found that mothers who were concerned about child weight reported higher food restrictions, while there was no statistically significant association between concern about child weight and pressure to eat. However, some studies did not demonstrate such associations [24,25,26]. For example, a study in the US (*n* = 196) reported no statistically significant association between concern about child weight and feeding practices involving restriction of food and pressure to eat [24]. Overall, despite the progressive evidence linking caregivers’ concern about child weight to their non-responsive feeding practices, current findings of the relationships between caregivers’ concern about child weight and their non-responsive feeding practices have been inconsistent.

In addition to the evidence above, some studies have suggested that caregivers’ concern about child underweight and overweight may have different effects on their non-responsive feeding practices [27,28,29]. For example, a cross-sectional study in Brazil (*n* = 659) reported that parental concern about child overweight was associated with more food restriction for children’s weight control and health, while concern about child underweight was associated with more pressure to eat [28]. Similarly, Gebru et.al [29] used a multi-stage random sampling method and found that caregivers who were concerned about child underweight were more likely to pressure their children to eat and might not restrict their children’s food, whereas caregivers who were concerned about child overweight were more likely to restrict their children’s food intake and less likely to force children to eat. Warkentin et.al [30] also found that parental higher concern about child overweight was associated with restrictive feeding, but not associated with practices such as forcing children to eat or using food as a reward after controlling for the confounders. Although the above findings supported that concern about child overweight and underweight had varied influences on different types of caregivers’ non-responsive feeding practices, no previous systematic reviews have pooled the results of various studies to verify their relationship. Thus, there is a need to synthesize the current evidence to identify the influence of caregivers’ concern about child underweight and overweight on their non-responsive feeding practices.

To sum up, current findings about the relationships between caregivers’ concern about child weight and their non-responsive feeding practices were equivocal. This systematic review and meta-analysis aimed to summarize the existing evidence on their associations. To the best of our knowledge, no systematic review and meta-analysis has been conducted to summarize the relationships between caregivers’ concern about child weight (including underweight and overweight) and their non-responsive feeding practices. Findings from this review will enhance our understanding of their relationships and inform the development of future interventions to optimize caregivers’ feeding practices. Furthermore, our review aimed to clarify the impacts of caregivers’ concern about child underweight and overweight on their non-responsive feeding practices. The findings will help to identify their impacts on caregivers’ non-responsive feeding practices, which may provide guidance on developing personalized interventions to improve caregivers’ feeding practices and eventually manage childhood obesity.

## 2. Methods

### 2.1. Data Sources and Search Strategy

The systematic review and meta-analysis complied with the guidelines of the Preferred Reporting Items for Systematic Reviews and Meta-Analysis (PRISMA) [31] and the Meta-analysis of Observational Studies in Epidemiology (MOOSE) [32], and was registered in PROSPERO (registration number: CRD42022304697). The PECO framework was used to formulate the question to explore the associations between the exposures and outcomes of interest [33].

A systematic literature search was carried out on PubMed, Embase, PsycINFO, Web of Science core collection, and CINAHL, from their inception to March 2022. To minimize publication bias, we also searched for studies in grey literature sources including the Grey literature report (http://greylit.org/, accessed on 10 March 2022), and Open grey EU (http://opengrey.eu/, accessed on 10 March 2022). The search was limited to publications published in English. The free text and Medical Subject Headings (MeSH) terms used for the search included: child, preschool, school child, paediatric, pre-teen, caregiver, parent, grandparent, mother, father, guardian, perception, concern, recognition, weight, body size, body mass index, feeding practice, food parenting, and food control. A manual search of the bibliography of the included studies was performed to identify additional studies.

### 2.2. Inclusion and Exclusion Criteria

Studies were included if they met the following criteria:(1)Study design was cohort, case-control, or cross-sectional study.(2)Studies that examined the relationships between caregivers’ concern about child weight and their non-responsive feeding practices.(3)The exposure was caregivers’ concern about child weight, including underweight and overweight.(4)The outcomes were caregivers’ non-responsive feeding practices.(5)Included caregivers (e.g., parents and grandparents) who were responsible for the food environment and their children’s eating.(6)Children aged 1 to 11 years at baseline (from toddler to middle childhood) [34]. Evidence showed that it is a critical period for the development of children’s self-regulation [35,36] and is characterised by growing independence from eating [37].

Studies were excluded if they:(1)Were reviews, editorials, commentaries, letters, or methodological papers.(2)Were non-English papers.(3)Did not report the statistics for the relationships between caregivers’ concern about their children’s weight and their non-responsive feeding practices.(4)Focused on children with diseases that might influence their eating.

### 2.3. Study Screening and Data Extraction

The PRISMA flow diagram was followed during the study screening stage [31]. One investigator (JW) screened the title and abstract for initial inclusion. Full texts were reviewed independently by two investigators (JW and XW) for further screening. Extracted data were compared and summarized to have one final document with which analysis was conducted. The information extracted included: the name of the first author, year of publication, the country that the study was conducted in, study design, sample size, response rate, variables of interest and their measures, and main findings. Data on the associations between caregiver’s concern about child weight and non-responsive feeding practices, including regression coefficients (*β*) or risk ratio (OR) and their 95% confidence intervals (CI) or standard errors were extracted. We contacted the corresponding authors if a study did not provide necessary numerical results. For any disagreements that occurred during the study screening and data extraction stages between the two investigators, a third reviewer (YC) was consulted.

### 2.4. Outcomes

Based on the conceptual analyses of food parenting practices [38,39,40], the non-responsive feeding practices were classified into four categories including restriction, pressure to eat, emotional feeding, and use of food as a reward.

(1)Restriction means that the caregivers enforce strict limitations on children’s access to food or opportunities to consume a specific food [15]. Typically, restrictive feeding is used to control the children’s intake of unhealthy food [19,41,42,43] and children’s weight [42].(2)Pressure to eat means that caregivers insist, demand, or physically struggle with the child to have the child eat enough or enough of a specific food [15,19,42,43].(3)Using food as a reward is also called instrumental feeding, which bribes and threatens children to eat food [15,44].(4)Emotional feeding means that caregivers use food to manage or calm children when they are upset, fussy, angry, hurt, or bored [15,45], such as by using food to soothe, intrigue, and/or relieve.

### 2.5. Study Quality Assessment

The Joanna Briggs Institute (JBI) Critical Appraisal Checklist for Analytical Cross-Sectional Studies and JBI Critical Appraisal Checklist for Cohort Studies were used for quality appraisal [46]. The tools assess the methodological quality of a study and determine the extent to which a study has addressed the possibility of bias in its design, conduct, and analysis. Two reviewers (JW and XW) independently performed the assessment, checking for possible sources of bias, attrition, and the validity of survey instruments. The final assessment was achieved upon discussion, and no studies were identified for exclusion by reviewers (see Appendix A).

### 2.6. Statistical Analysis

The studies eventually without necessary numerical results were excluded from the meta-analysis. We thus used a semi-quantitative approach to summarize the findings of all the included articles as adopted by recent reviews [47,48,49].

The studies that provided the necessary numerical results were included in the meta-analysis. If an association between the same category of exposure and outcome was multiply evaluated in one study, the results were first synthesized within the study, and the summarized data were then used for the meta-analysis. The subscale of concern about child weight in the Child Feeding Questionnaire (CFQ) was used to assess caregivers’ concern about the child’s risk of being overweight [19], which was recategorized as concern about child overweight in the meta-analysis. All extracted effect sizes (*β* or OR) were adjusted values from multivariable models in each article that examined caregivers’ concern about child overweight and their non-responsive feeding practices and were pooled in the meta-analysis. Due to the limited number of articles that assessed the associations between caregivers’ concern about child underweight and their pressure to make children eat, we decided to include a study [50] that did not control for the confounders in the meta-analysis. The heterogeneity of the included studies was investigated using the *I*^2^ statistics [51]. The random-effects model was used in case of high heterogeneity indicated by an *I*^2^ > 50%; otherwise, the fixed-effects model was used [52,53]. The potential publication bias was assessed by the combination of Egger’s test and visual inspection of the funnel plot [54]. The leave-one-out (LOO) analysis was also performed as the sensitivity analysis to investigate the influence of a single study on the pooled effect [55]. Additionally, differences caused by child age [56,57], child weight status [58], caregivers’ education [30,56], caregivers’ role [59], family income [60,61], country’s level of development [62,63], and the measurements of exposure and outcome [19,64] were evaluated using subgroup analysis and meta-regression. All analyses were performed in Stata 17.0 (StataCorp, College Station, TX, USA). All tests were two-sided, and the statistical significance was set as a *p*-value < 0.05.

## 3. Results

### 3.1. Search Results

A total of 35,780 articles were identified. After removing duplicates, 23,194 articles remained for the initial screening, from which 239 articles were retrieved. After full text screenings, 35 studies were retained for analyses. The PRISMA flow diagram is shown in Figure 1.

### 3.2. Characteristics of the Studies

Characteristics of the included studies are shown in Table 1. The studies were published between 2001 and 2022, conducted in the US (*n* = 12) [21,24,65,66,67,68,69,70,71,72,73,74], Australia (*n* = 6) [26,27,75,76,77,78], Brazil (*n* = 4) [28,30,50,79], Sweden (*n* = 4) [22,80,81,82], China (*n* = 2) [83,84], the Netherlands (*n* = 1) [85], Ethiopia (*n* = 1) [29], the UK (*n* = 1) [86], Mexico (*n* = 1) [87], South Korea (*n* = 1) [88], Portugal (*n* = 1) [23], India (*n* = 1) [25], and France (*n* = 1) [74]. The study designs were a cross-sectional study (*n* = 32) and a cohort study (*n* = 3) [23,26,85]. The total sample size was 22,933, with the individual study sample size ranging from 48 [26] to 4689 [85]. The caregivers were typically mothers (*n* = 19). Only four studies used a random sampling method [27,29,67,83].

### 3.3. Measurements for Caregivers’ Concern about Child Weight and Their Non-Responsive Feeding Practices

The measurements used for assessing caregivers’ concerns about child weight and their non-responsive feeding practices are shown in Appendix A. The most common measurement was CFQ [19], which was intended for parents of children aged 2–11 years. Thirty studies used the subscale of concern about child weight in CFQ to assess caregivers’ concern about children at risk of being overweight [19], and two studies [29,77] used the Preschooler Feeding Questionnaire (PFQ) [89] to assess caregivers’ concern about child underweight. CFQ (*n* = 21) and the Chinese Child Feeding Questionnaire (C-CFQ) [90] (*n* = 2) [83,84] was used to assess the domains of non-responsive feeding practices, including restriction, pressure to eat, and use of food as a reward. The Comprehensive Feeding Practices Questionnaire (CFPQ) [64] was also adopted in ten studies to assess caregivers’ restrictions for weight and health, pressure to eat, use of food as a reward, and emotional feeding.

### 3.4. Semi-Quantitative Results

Table 2 summarizes the associations between caregivers’ concerns about child weight and their non-responsive feeding practices. The included studies focused on examining the relationships between caregivers’ concern about child weight and their restrictive feeding and pressure to eat. Specifically, thirty-four studies examined the relationships between caregivers’ concern about child weight and restrictive feeding practice, and 40 statistically significant associations were observed. A total of 34 statistical estimates of the relationships between caregivers’ concern about child weight and pressure to eat were investigated in 24 studies, with over one-third of the associations being statistically significant. A detailed summary of the associations is reported in Appendix A.

### 3.5. Results of Meta-Analysis

#### 3.5.1. Concern about Child Overweight and Restrictive Feeding

Figure 2 shows that caregivers’ excessive concern about child overweight was associated with more use of restrictive feeding, with a pooled *β =* 0.22 (95%CI: 0.12, 0.31) based on the random-effects model (*I*^2^ = 95%, *p* < 0.001). The funnel plot of *β* for these studies appears symmetric (Figure 3). No statistically significant publication bias was detected (Egger’s test *p* = 0.106). The LOO sensitivity analysis indicated that the results were robust, and all the estimates were statistically significant (Figure 4).

Compared with the caregivers who were not concerned about child overweight, the caregivers who were concerned were found to use restrictive feeding more frequently (pooled OR = 2.34; 95%CI: 1.69, 3.23; Figure 5) by using the random-effects model. The funnel plot of ORs appears symmetric (Figure 6). No statistically significant publication bias was found (Egger’s test *p* = 0.473). The LOO analysis indicated that the results were robust, and all the estimates were statistically significant (Figure 7).

#### 3.5.2. Concern about Child Weight and Pressure to Eat

As shown in Figure 8, caregivers’ concern about child overweight was not statistically significantly associated with pressure to eat (pooled *β* = −0.05; 95%CI: −0.13, 0.04), using the random-effects model (*I*^2^ = 79.1%, *p* < 0.001). The funnel plot of *β*s for these studies appears symmetric (Figure 9). No statistically significant publication bias was found (Egger’s test *p* = 0.670). The LOO analysis indicated that the results were robust, and only one estimate was statistically significant (Figure 10).

Figure 11 presents the pooled ORs for the effects of concern about child underweight and overweight on pressure to eat. Caregivers’ concern about child underweight was associated with increased risk of pressure to eat (OR = 1.83; 95%CI:1.44, 2.33) compared with those who were not concerned about underweight. In contrast, the result showed that the caregivers who were concerned about child overweight were less likely to force their children to eat (OR = 0.76; 95%CI: 0.59, 0.98) compared with those who were not concerned. As there were no more than three studies exploring their associations, funnel plots were not provided.

#### 3.5.3. Concern about Child Overweight and Use of Food as a Reward

Figure 12 shows that more caregivers’ concern about child overweight was associated with less use of food as a reward (*β* = −0.06; 95%CI: −0.11, −0.01) with the fixed-effects model (*I*^2^ = 0.0%, *p* = 0.852). The funnel plot of *β*s for these studies appears symmetric (Figure 13). No statistically significant publication bias was found (Egger’s test *p* = 0.943). The LOO analysis indicated that the results were robust, and only one estimate was not statistically significant (Figure 14).

#### 3.5.4. Subgroup Analysis

The subgroup analysis aimed to identify the potential sources of heterogeneity and examine the stability of the relationship between caregivers’ concern about child weight and their non-responsive feeding practices across different categories. The results indicated that the associations were similar in most subgroups (Table 3). However, moderate to high heterogeneities were observed in most subgroups. The increased likelihood of restrictive feeding was found in the subgroup with less high-income families (*β* = 0.320; 95% CI: 0.233, 0.408; *n* = 3). No statistically significant between-group differences were found for other subgroup comparisons (Table 3).

## 4. Discussion

The aim of this systematic review and meta-analysis was to summarize the available evidence examining the relationships between caregivers’ concern about child weight and their non-responsive feeding practices. The results suggested that caregivers’ concern about child weight may play a significant role in the use of non-responsive feeding practices, especially restrictive feeding.

Thirty-four studies examined 52 relationships between caregivers’ concern about child weight and their restrictive feeding with 40 statistically significant associations being observed in the semi-quantitative analyses. This finding suggested the important role of concern about child weight in restrictive feeding, which was in accordance with Costanzo and Woody’s suggestion [20] that caregivers may use controlled feeding practices when they are concerned about their children’s weight. Consistently, the results of this meta-analysis indicated that caregivers’ concern about child overweight was associated with more restriction of food. When caregivers are concerned about child overweight, they may realize the consequences of child overweight or obesity [2,3,4], such as adulthood obesity and chronic disease, which may make them adopt controlled feeding practices (e.g., the restriction of unhealthy food) [83]. Furthermore, society places great importance on weight and body shape [91]; thus, caregivers may be more likely to restrict their children’s eating if they were concerned about their child becoming overweight. However, restrictive feeding practices have been linked with increased disinhibited eating and weight gain among children [71,72,86], hence current obesity prevention and treatment guidelines recommend that caregivers avoid excessive restriction of children’s eating [92].

Of twenty-four studies examining the associations between caregivers’ concern about child weight and the use of pressure to eat, over one third of the associations were statistically significant in the semi-quantitative results. This finding suggested that caregivers’ concern about child weight may be a potential risk factor of pressure to eat. However, the results of our meta-analysis were mixed. Specifically, the pooled *β*s showed that caregivers’ concern about child overweight was not associated with pressure to eat, but the pooled ORs presented that caregivers’ concern about both child underweight and overweight were statistically significantly associated with pressure to eat. That is, caregivers who were concerned about child underweight may be more likely to apply pressure to eat compared to those who were not concerned. In contrast, caregivers who were concerned about child overweight were less likely to force their children to eat compared with those who were not concerned. Caregivers might consider that a lower weight, which may be a biological (heritable) characteristic of the child, could compromise their healthy development and growth [93]. Thus, they may directly force children to eat more in response to their concern about their child being underweight. Nevertheless, caregivers’ pressure to eat has been associated with negative affective reactions to food and a close relationship with low weight in children [94]. Additionally, we found the pooled *β*s and ORs that synthesized associations between caregivers’ concern about child overweight and their pressure to eat were inconsistent. It suggests that concern about child overweight may not be the main reason for the low frequency of caregivers’ pressure to eat. Children may have some difficulties in developing optimal eating habits due to their physical and psychological characteristics (e.g., limited autonomy) [69,95,96]. In this case, caregivers may assume that children are incapable of detecting their own cues to hunger and satiety, and consequently, they may prefer to manage their eating through pressure [97]. Furthermore, other potential risk factors (e.g., perception of child weight, child fussy eating) may be more directly associated with caregivers’ pressure to eat [98,99]. Due to the limited number of included studies, the relationships between concern about child overweight and pressure to eat need to be investigated further.

Six studies reported the relationships between caregivers’ concern about child overweight and their use of food as a reward. These studies examined a total of 11 associations and only two showed statistical significance in the semi-quantitative analyses, suggesting their weak significance. The result of the meta-analysis showed that caregivers’ excessive concern about child overweight was negatively associated with use of food as a reward. Caregivers’ feeding practices may carry cultural and ethnic variations. In developing countries (e.g., China) or in low socioeconomic settings, caregivers tend to use food as a symbol of their love for their children, or as an educational and emotional tool for shaping their children’s eating and behaviors [62], because they may believe that higher weight indicates better health and nutrition status [100,101]. However, if they expressed concern about their child being overweight, they may not use food as a reward to encourage their children to eat more [84].

In addition, the heterogeneities were high in the meta-analysis that synthesized the relationships between caregivers’ concern about child overweight and restrictive feeding and pressure to eat with pooled *β*s. First, our subgroup analysis showed that caregivers who were concerned about child overweight may be more likely to restrict their children’s food in the group with less high-income families. Caregivers in low-income families may experience unique barriers to provide health-promoting feeding practices [100,102]. When they are concerned about child overweight, they may not consider applying responsive feeding practices (e.g., rules and limits, modeling, and the encouragement of healthy eating). Instead, they may prefer to use restrictive feeding as a straightforward method [83]. Second, although the subscale of concern about child weight in the CFQ has been commonly used [19], it has been criticized that it is not possible to know whether the concern was about either current or future overweight or a combination of both [70,86]. This might have a varied influence on caregivers’ non-responsive feeding practices and be one of the sources of heterogeneities. Third, the heterogeneities might be due to the inconsistent confounding adjustment in the included studies. Some covariates (e.g., child sex, child temperament, and caregivers’ perception of child weight) had close links to caregivers’ non-responsive feeding practices [88,98]. However, some studies did not include these variables in the multivariate analysis, which might have contributed to the inconsistent findings and thus further affect the pooled estimates. Therefore, further studies should consider the confounding factors thoroughly when examining these relationships.

### 4.1. Limitations and Strengths

To the best of our knowledge, this is the first review that comprehensively synthesized data on the relationships between caregivers’ concern about child weight and their non-responsive feeding practices. Our review included rigorous methodological procedures to obtain and pool data from 22,933 respondents. We also adopted a wide range of search terms to retrieve all potential articles published in English, including the grey literature. Furthermore, we used search terms indicating wider age ranges than our target age range to avoid missing relevant articles by omission. In addition, all estimates we provided for the relationships between caregivers’ concern about child overweight and their non-responsive feeding practices were adjusted for the covariates in the meta-analysis. However, there are several limitations to this systematic review and meta-analysis. First, some studies did not report a standard error or 95%CI, which precluded us from pooling all the extracted data to examine the associations. Thus, the results from the meta-analysis should be interpreted with caution. Second, most studies with cross-sectional designs precluded us from establishing causal inferences. Third, all included studies employed self-reported questionnaires to assess our interest variables, which may be subject to recall bias. In addition, the included studies were mainly conducted in western countries such as the US and Australia. Findings from this systematic review might not be extrapolated to other populations (e.g., Asian).

### 4.2. Implications

More longitudinal studies are needed to better understand the impact of caregivers’ concern about child weight on their non-responsive feeding practices. Such studies should be adequately powered and include a representative sample. Second, validated instruments are required to assess caregivers’ concern about their children’s current and future weight separately. Third, more potential confounding factors (e.g., demographics, child temperament, and parental perception of child weight) should be taken into consideration for comprehensive analysis. Furthermore, it is essential for professionals and clinicians to provide guidance on how to manage and reduce caregivers’ concern about child weight, particularly among low-income families. It may help them implement appropriate feeding practices and eventually control their children’s weight.

## 5. Conclusions

This systematic review synthesized the evidence and indicated that caregivers’ concerns about child weight may be a significant risk factor for non-responsive feeding practices. Caregivers who are concerned about child overweight may be more likely to adopt restrictive feeding and less likely to use food as a reward. They might more frequently apply pressure to eat when they were concerned about child underweight. Given the consequences of non-responsive feeding practices and the role of caregivers’ concern about child weight on the use of such practices, interventions that manage and reduce caregivers’ excessive concern about child underweight or overweight may help to optimize feeding practices and eventually contribute to child health. Additionally, future prospective and experimental, theory-driven studies using validated measurements and representative sampling while controlling for potential covariates are needed to provide more evidence with regard to these causal relationships.

## Figures and Tables

**Figure 1 nutrients-14-02885-f001:**
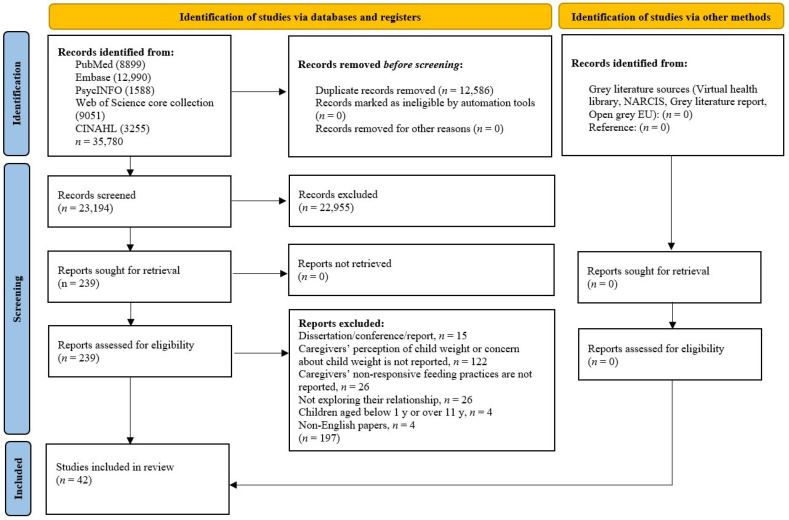
PRISMA flow diagram for screening and selection of articles.

**Figure 2 nutrients-14-02885-f002:**
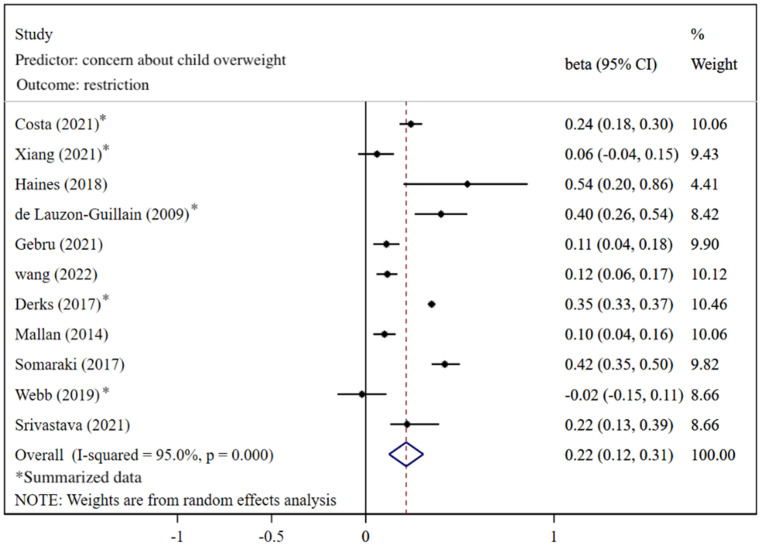
Effects of concern about child overweight on restrictive feeding (evaluated using *β*) [21,22,23,26,27,29,74,76,83,84,85].

**Figure 3 nutrients-14-02885-f003:**
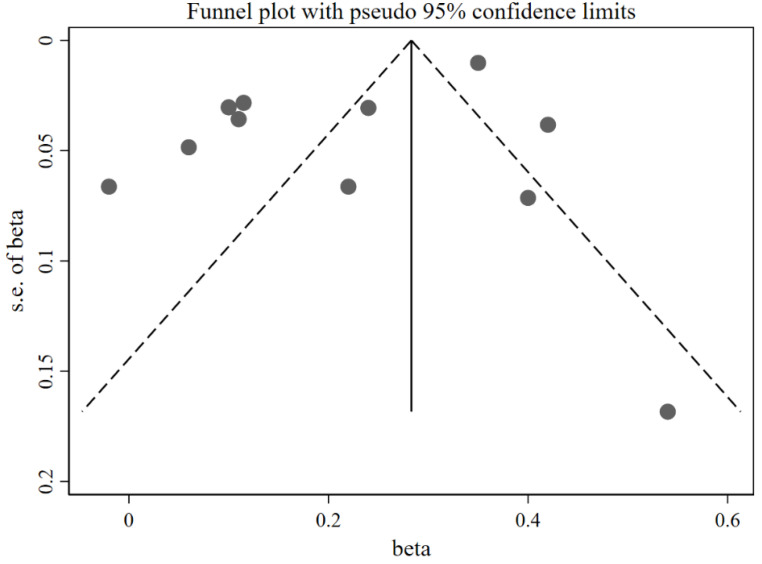
Funnel plot with pseudo 95% confidence limit for the studies exploring the effects of concern about child overweight on restrictive feeding (evaluated using *β*).

**Figure 4 nutrients-14-02885-f004:**
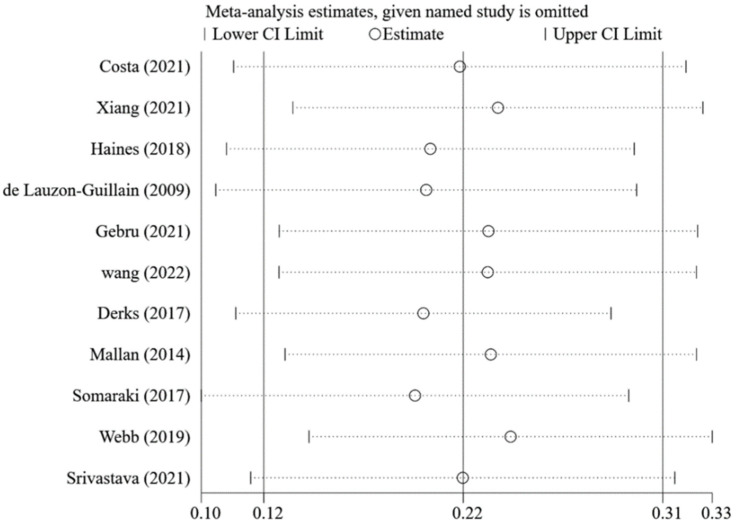
Pooled *β*s of the associations between concern about child overweight and restrictive feeding in leave-one-out analysis [21,22,23,26,27,29,74,76,83,84,85].

**Figure 5 nutrients-14-02885-f005:**
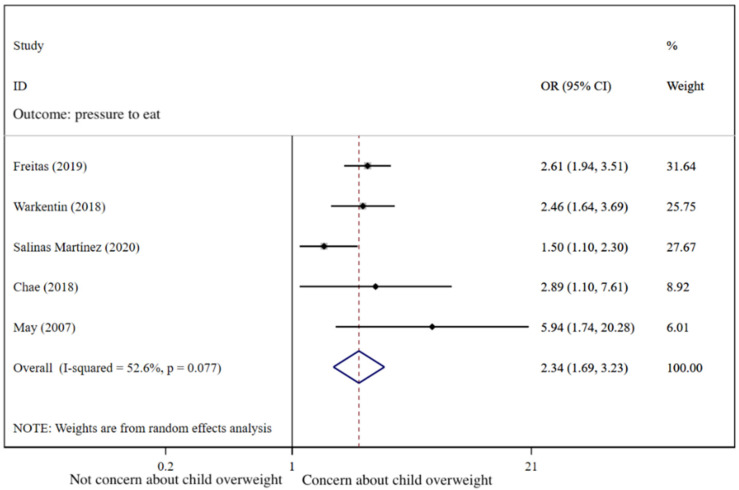
Effects of concern about child overweight on restrictive feeding (evaluated using OR) [30,71,79,87,88].

**Figure 6 nutrients-14-02885-f006:**
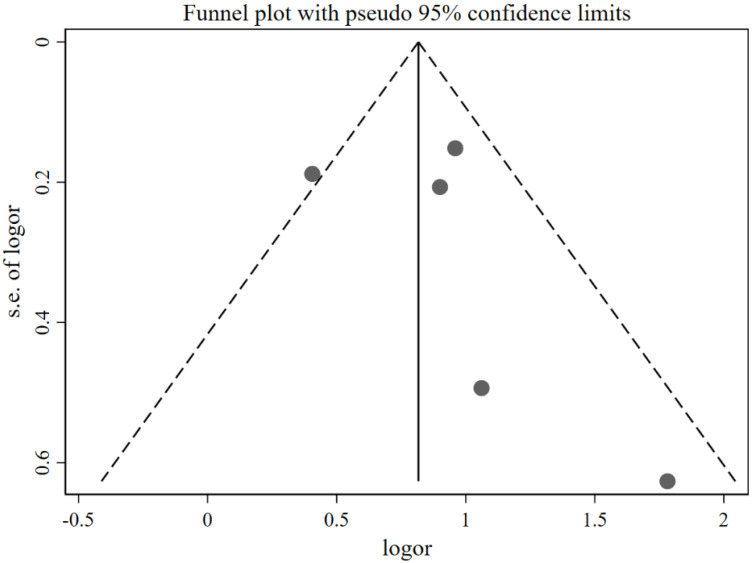
Funnel plot with pseudo 95% confidence limit for the studies exploring the effects of concern about child overweight on restrictive feeding (evaluated using OR).

**Figure 7 nutrients-14-02885-f007:**
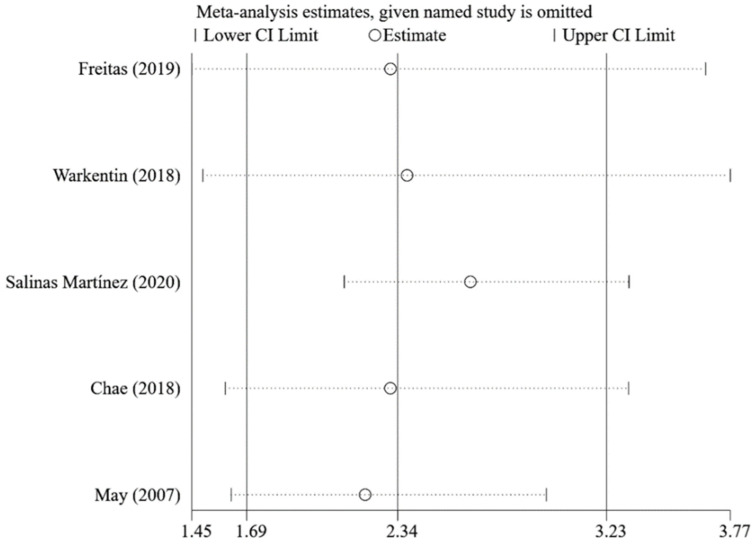
Pooled ORs for restrictive feeding of the concern about overweight group vs. non-concern about overweight group in leave-one-out analysis [30,71,79,87,88].

**Figure 8 nutrients-14-02885-f008:**
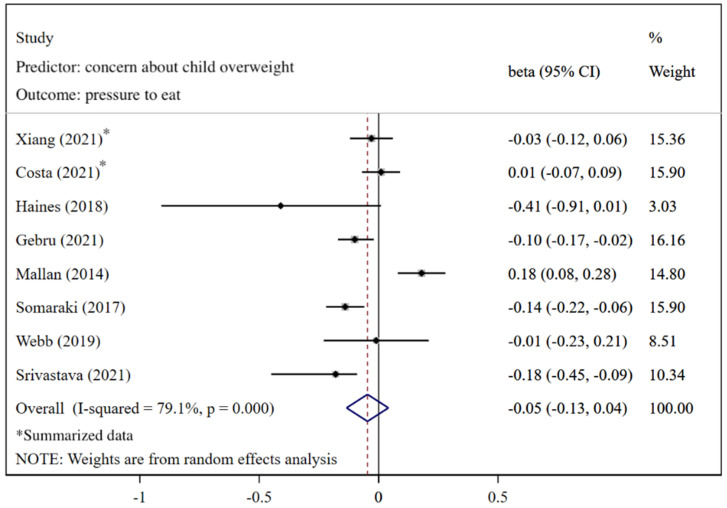
Effects of concern about child overweight on pressure to eat (evaluated using *β*) [21,22,23,26,27,29,76,83].

**Figure 9 nutrients-14-02885-f009:**
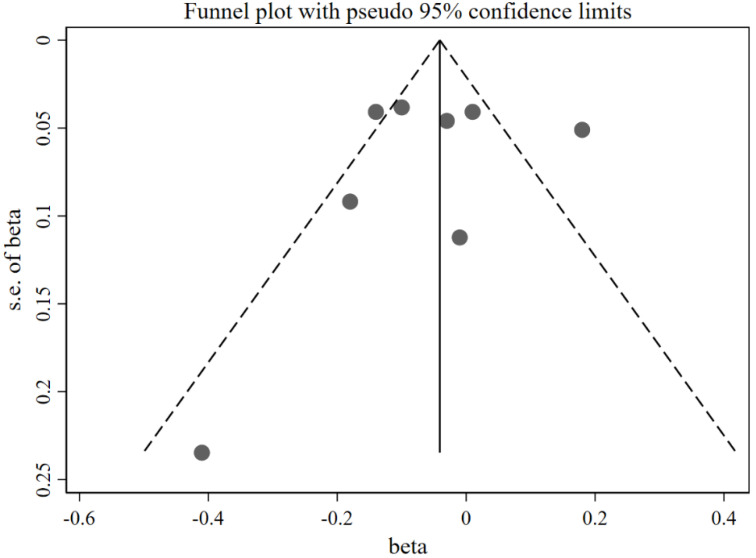
Funnel plot with pseudo 95% confidence limit for the studies exploring the effects of concern about child overweight on pressure to eat (evaluated using *β*).

**Figure 10 nutrients-14-02885-f010:**
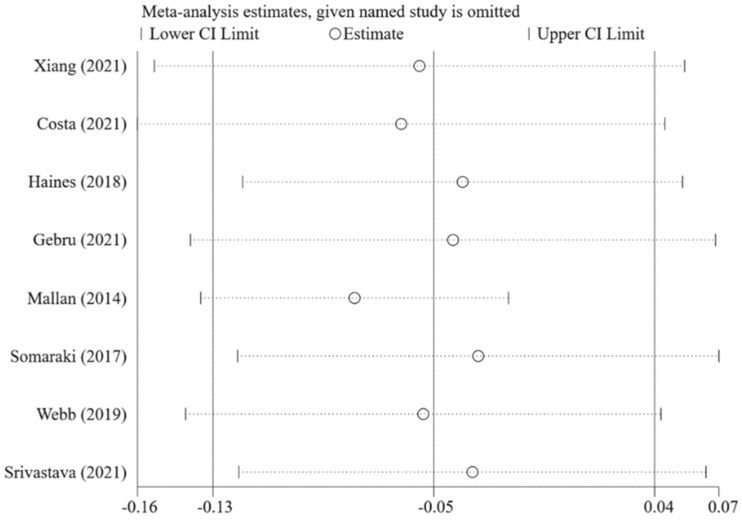
Pooled *β*s of the associations between concern about child overweight and pressure to eat in leave-one-out analysis [21,22,23,26,27,29,76,83].

**Figure 11 nutrients-14-02885-f011:**
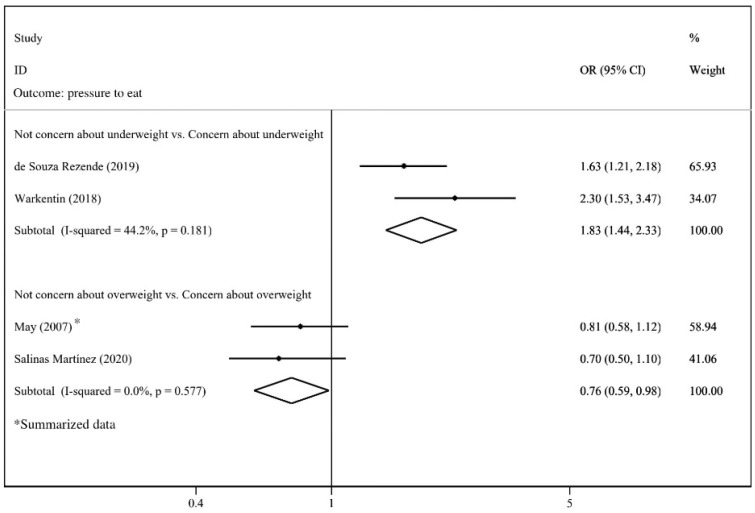
Effects of concern about child overweight/underweight on pressure to eat (evaluated using OR) [30,50,71,87].

**Figure 12 nutrients-14-02885-f012:**
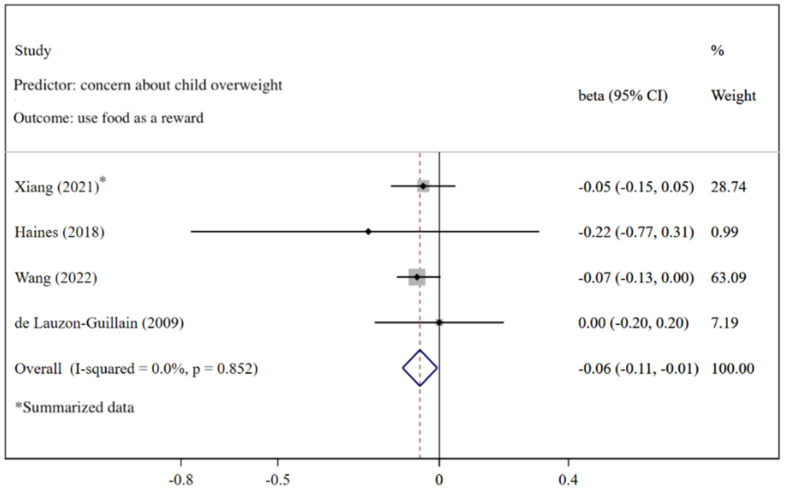
Effects of concern about child overweight on use of food as a reward (evaluated using *β*) [27,74,83,84].

**Figure 13 nutrients-14-02885-f013:**
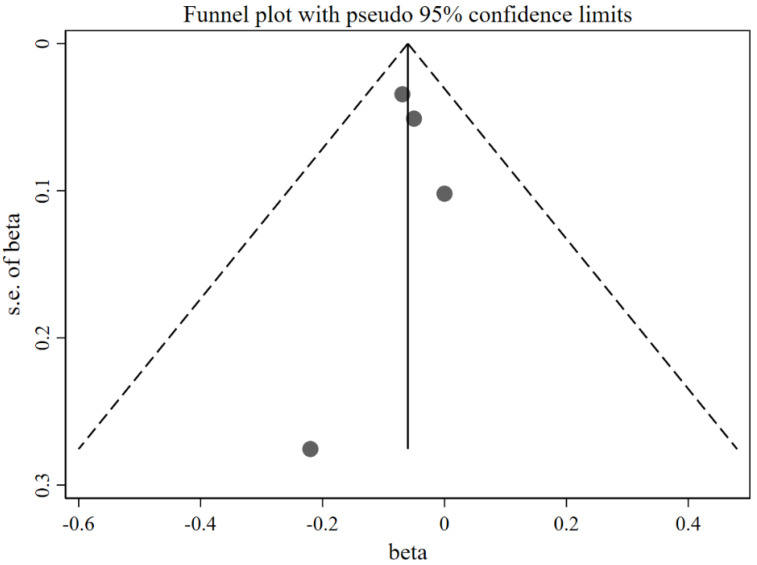
Funnel plot with pseudo 95% confidence limit for the studies exploring the effects of concern about child overweight on use of food as a reward (evaluated using *β*).

**Figure 14 nutrients-14-02885-f014:**
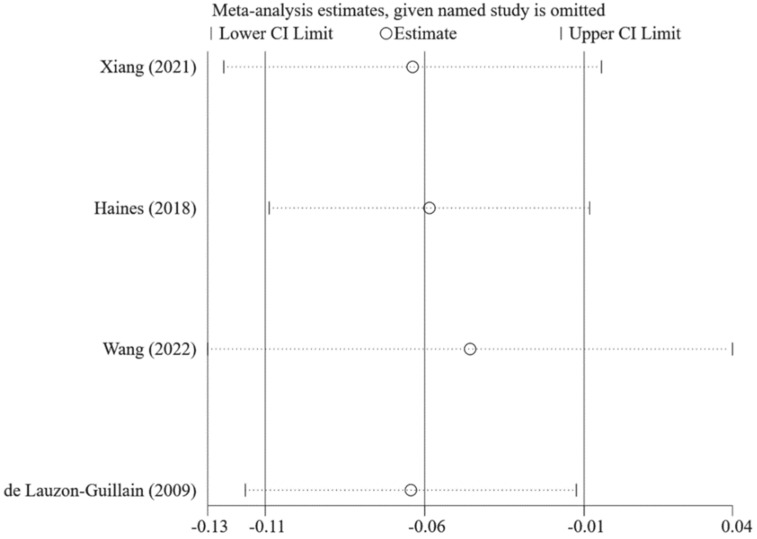
Pooled *β*s of the associations between concern about child overweight and use of food as a reward in leave-one-out analysis [27,74,84,87].

**Table 1 nutrients-14-02885-t001:** Characteristics of the included studies (*n* = 35).

First Author, Year	Country	Study Design	Caregivers	Age of Children	Sampling Method	Sample Size	Response Rate
Xiang, 2021 [83]	China	Cross-sectional study	Parents	4.54 ± 0.85 years	Random cluster sampling	1616	100%
Branch, 2017 [65]	US	Cross-sectional study	Mothers	5.39 ± 0.75 years	Voluntary (response) sample	264	66.50% (264/397)
Francis, 2001 [24]	US	Cross-sectional study	Mothers	5.4 ± 0.02 years	Voluntary (response) sample	196	99.49% (196/197)
Freitas, 2019 [79]	Brazil	Cross-sectional study	Mothers	2–8 years	Voluntary (response) sample	835	70.88% (835/1178)
Webber, 2010 [86]	UK	Cross-sectional study	Mothers	8.3 ± 0.63 years	Voluntary (response) sample	213	52.59% 213/405
Gebru, 2021 [29]	Ethiopia	Cross-sectional study	Caregivers (mother/father/grandmother and other)	4.5 ± 0.04 years	Multi-stage random sampling	525	96.86% (525/542)
de Souza Rezende, 2019 [50]	Brazil	Cross-sectional study	Mothers	4.98 ± 1.8 years	Voluntary (response) sample	927	78.69% (927/1178)
Mais, 2017 [28]	Brazil	Cross-sectional study	Parents	5–9 years	Secondary data	659	46.08% (659/1430)
Cachelin, 2013 [66]	US	Cross-sectional study	Mothers	2–11 years	Voluntary (response) sample	425	75.49% (425/563)
Ek, 2016 [80]	Sweden	Cross-sectional study	Parents	5.5 ± 1.0 years	Representative sample	478	51.34% (478/931)
Derks, 2017 [85]	Netherlands	Longitudinal study (cross-sectional relationship)	Parents	9.76 ± 0.29 years	Voluntary (response) sample	4689	54.85% (4689/8548)
Eli, 2016 [81]	Sweden	Cross-sectional study	Mothers	4.5 ± 0.4 years	Voluntary (response) sample	876	29.13% (876/3007)
Gregory, 2010 [77]	Australia	Cross-sectional study	Mothers	3.3 ± 0.84 years	Voluntary (response) sample	183	100% (183/183)
Wang, 2022 [84]	China	Cross-sectional study	Mothers	4.56 ± 1.35 years	Convenience sample	1106	95.02% (1106/1164)
Haines, 2018 [27]	Australia	Cross-sectional study	Mothers	5.0 ± 0.1 years	Random sample	310	58.71% (310/528)
Bouhlal, 2018 [67]	US	Cross-sectional study	Mothers	4–5 years	Random sample	221	100%
de Lauzon-Guillain, 2009 [74]	US and France	Cross-sectional study	Parents	3.7–6.8 years	Not clear	219	100%
Srivastava, 2021 [21]	US	Cross-sectional study	Parents	3.95 ± 0.75 years	Voluntary (response) sample	273	31.67% (273/862)
Brann, 2010 [68]	US	Cross-sectional study	Caregivers	4.5 ± 1.5 years	Voluntary (response) sample	123	41.28% (123/298)
Webb, 2019 [26]	Australia	Longitudinal study	Parents	7.6 ± 0.8 years	Voluntary (response) sample	48	100%
Warkentin, 2018 [30]	Brazil	Cross-sectional study	Parents	2-5 years	Voluntary (response) sample	402	40.36% (402/996)
Tan, 2011 [69]	US	Cross-sectional study	Parents	3–9 years (mean age 5.6 years)	Voluntary (response) sample	63	100%
Somaraki, 2017 [22]	Sweden	Cross-sectional study	Mothers	4-7 years (mean age 4.8 years)	Population sample	1284	96.91% (1284/1325)
Salinas Martínez, 2020 [87]	Mexico	Cross-sectional study	Mothers	4.0 ± 1.2 years	Consecutive selection sample	507	100%
Rodgers, 2013 [75]	Australia	Cross-sectional study	Mothers	2.03 ± 0.37 years	Voluntary (response) sample	218	99.09% (218/220)
Loth, 2021 [70]	US	Cross-sectional study	Parents	6.4 ± 0.8 years	Voluntary (response) sample	149	149/150 (99.33%)
Mallan, 2014 [76]	Australia	Cross-sectional study	Fathers	3.5 ± 0.9 years	Voluntary (response) sample	342	78.44% (342/436)
Chae, 2018 [88]	South Korea	Cross-sectional study	Mothers	3–5 years	Convenience sample	223	100%
Costa, 2021 [23]	Portugal	Longitudinal study (cross-sectional relationship)	Mothers	4–7 years (4 years at the baseline)	National population sample	3233	38.73% (3233/8647)
May, 2007 [71]	US	Cross-sectional study	Mothers	24–59 months	Voluntary (response) sample	967	73.87% (967/1309)
Seburg, 2014 [72]	US	Cross-sectional study	Parents	6.6 ± 1.7 years	Voluntary (response) sample	391	92.87% (391/421)
Crouch, 2007 [78]	Australia	Cross-sectional study	Mothers	4.42 ± 1.35 years	Voluntary (response) sample	111	99.11% (111/112)
Ayine, 2020 [73]	US	Cross-sectional study	Parents	6–10 years (mean age 8.42 years)	Voluntary (response) sample	169	100%
Jani Mehta, 2014 [25]	India	Cross-sectional study	Mothers	34 ± 14 months	Convenience sample	203	88.26% (203/230)
Nowicka, 2014 [82]	Sweden	Cross-sectional study	Parents	4.5 ± 0.3 years	National population sample	564	18.76% (564/3007)

**Table 2 nutrients-14-02885-t002:** Summary of the associations between caregivers’ concern about child weight and their non-responsive feeding practices (*n* = 35).

First Author, Year	Concern about Child Weight → Non-Responsive Feeding Practices			
	Restriction	Pressure to Eat	Food as a Reward	Emotional Feeding
Ayine, 2020 [73]	+ ^a,1^	Φ ^a,1^		
Francis, 2001 [24]	Φ ^a,2^	Φ ^a,2^		
Freitas, 2019 [79]	+ ^b,2^			
Webber, 2010 [86]	+ ^c,2^	Φ ^c,2^		
Loth, 2021 [70]	+ ^c,2^	Φ ^c,2^		
May, 2007 [71]	+ ^b,2^	Φ ^b,2^ (PEA) − ^b,2^ (PEE) Φ ^b,2^ (PER)		
Gebru, 2021 [29]	+ ^e,2^ Φ ^d,2^	− ^e,2^ + ^d,2^		
Crouch, 2007 [78]	+ ^a,2^			
de Souza Rezende, 2019 [50]		+ ^g,1^		
Jani Mehta, 2014 [25]	Φ ^f,1^	Φ ^f,1^		
Salinas Martínez, 2020 [87]	+ ^b,2^	Φ ^b,2^		Φ ^b,2^
Mais, 2017 [28]	+ ^b,2^ (RFW) + ^b,2^ (RFH) Φ ^g,2^ (RFW) Φ ^g,2^ (RFH)	Φ ^b,2^ + ^g,2^	Φ ^b,2^ Φ ^g,2^	
Nowicka, 2014 [82]	+ ^a,2^			
Costa, 2021 [23]	+ ^b,2^ (4 y) + ^b,2^ (7 y)	Φ ^b,2^ (4 y) Φ ^b,2^ (7 y)		
Wang 2022 [84]	+ ^e,2^		− ^e,2^	
Xiang, 2021 [83]	− ^b,2^ (underweight children) + ^b,2^ (normal weight children) Φ ^b,2^ (overweight children)	Φ ^b,2^ (underweight children) Φ ^b,2^ (normal weight children) Φ ^b,2^ (overweight children)	Φ ^b,2^ (underweight children) Φ ^b,2^ (normal weight children) Φ ^b,2^ (overweight children)	
Branch, 2017 [65]	+ ^c,2^	Φ ^c,2^		
Cachelin, 2013 [66]	+ ^a,2^ (Hispanic model) + ^a,2^ (White model)			
Ek, 2016 [80]	+ ^a,2^	Φ ^a,2^		
Derks, 2017 [85]	+ ^e,2^ (model 1) + ^e,2^ (model 2) + ^e,2^ (model 3)			
Eli, 2016 [81]	+ ^a,2^	Φ ^a,2^		
Gregory, 2010 [77]	+ ^e,2^ Φ ^d,2^	Φ ^e,2^ + ^d,2^		
Haines, 2018 [27]	+ ^b,2^ + ^g,2^	Φ ^b,2^ + ^g,2^	Φ ^b,2^ + ^g,2^	Φ ^b,2^ Φ ^g,2^
Bouhlal, 2018 [67]	+ ^a,2^			
de Lauzon-Guillain, 2009 [74]	+ ^e,2^ (RFW) + ^e,2^ (RFH)		Φ ^e,2^	Φ ^e,2^
Srivastava, 2021 [21]	+ ^a,2^	− ^a,2^		
Brann, 2010 [68]	+ ^a,1^	+ ^a,1^		
Webb, 2019 [26]	Φ ^a,2^ (RFW) Φ ^a,2^ (RFH)	Φ ^a,2^		
Warkentin, 2018 [30]	+ ^b,2^ (RFW) Φ ^b,2^ (RFH) Φ ^g,2^ (RFW) Φ ^g,2^ (RFH)	Φ ^b,2^ + ^g,2^	Φ ^b,2^ Φ ^g,2^	
Tan, 2011 [69]	+ ^a,1^ (RFW) + ^a,1^ (RFH)			
Somaraki, 2017 [22]	+ ^a,2^	− ^a,2^		
Rodgers, 2013 [75]	+ ^a,2^			
Mallan, 2014 [76]	+ ^a,2^	+ ^a,2^		
Chae, 2018 [88]	+ ^b,2^			
Seburg, 2014 [72]	+ ^a,2^	Φ ^a,2^		
Number of significant associations	40	12	2	0
Total number of tested associations	52	34	11	4
Number of articles	34	24	6	3

NOTES. ^a^ Predictor: Concern about child weight (CFQ, continuous variable); ^b^ Predictor: Not concern about child overweight vs. Concern about child overweight; ^c^ Predictor: Not concern about child overweight vs. Little concern about child overweight vs. Concern about child overweight; ^d^ Predictor: Concern about child underweight (continuous variable); ^e^ Predictor: Concern about child overweight (continuous variable); ^f^ Predictor: Not concern about child weight vs. Concern about child weight; ^g^ Predictor: Not concern about child underweight vs. Concern about child underweight. ^1^ Not control for covariates; ^2^ Control for covariates. +, positive and statistically significant association between predictor and outcome; −, negative and statistically significant association between predictor and outcome; Φ, statistically non-significant association between predictor and outcome. RFW: restriction for weight; RFH: restriction for health; PEA: Pressure to eat all; PEE: Pressure to eat enough; PER: Pressure to eat right food.

**Table 3 nutrients-14-02885-t003:** Subgroup analysis of the effects of caregivers’ concern about child weight and their feeding practices.

Exposure/Outcome	Eligible Studies	Effect Size	Effect Estimates (95% CI)	*p* Value for Heterogeneity	I^2^ (%)	*p* Value between Groups
**Concern about Child Overweight/Restriction**						
Outcome measure		beta				0.636
CFQ	8		0.203 (0.103, 0.303)	<0.001	96.0	
CFPQ	3		0.288 (−0.055, 0.631)	<0.001	91.3	
Exposure measure						0.740
Two/three items	6		0.201 (0.071, 0.331)	<0.001	96.1	
One item	5		0.223 (0.109, 0.337)	<0.001	86.3	
Country						0.220
Developed country	9		0.245 (0.148, 0.342)	<0.001	94.2	
Developing country	2		0.101 (0.053, 0.149)	0.327	0.0	
Sample size						0.978
<500	5		0.214 (0.062, 0.365)	<0.001	85.7	
≥500	6		0.218 (0.105, 0.331)	<0.001	96.2	
Children’s mean age						0.884
≤5 years old	7		0.193 (0.089, 0.298)	<0.001	90.7	
>5 years old	2		0.171 (-0.192, 0.533)	<0.001	96.7	
Child overweight/obesity						0.523
≤20%	4		0.262 (0.084, 0.439)	<0.001	94.3	
>20%	2		0.154 (-0.022, 0.330)	0.002	89.9	
Caregivers’ role						0.268
Mothers (only)	4		0.293 (0.137, 0.450)	<0.001	93.3	
Parents/grandparents/fathers	7		0.174 (0.045, 0.303)	<0.001	95.9	
Caregivers’ education						0.412
Less than half with college degree or higher	2		0.104 (0.059, 0.150)	0.831	0.0	
More than half with college degree or higher	7		0.228 (0.090, 0.366)	<0.001	91.8	
Family income						0.027
High-income percentage ^a^ below median level ^b^	3		0.320 (0.233, 0.408)	0.002	83.8	
High-income percentage ^a^ above median level ^b^	3		0.126 (0.079, 0.172)	0.309	14.9	
**Concern about Child Overweight/Pressure to Eat**						
Outcome measure		beta				0.613
CFQ	6		−0.038 (−0.132, 0.056)	<0.001	83.8	
CFPQ	2		−0.157 (−0.535, 0.221)	0.124	57.7	
Exposure measure						0.965
Two/three items	5		−0.048 (−0.182, 0.086)	<0.001	86.2	
One item	3		−0.023 (−0.112, 0.065)	0.191	39.6	
Country						0.880
Developed country	7		−0.052 (−0.158, 0.053)	<0.001	82.0	
Developing country	1		−0.030 (−0.120, 0.060)	/	/	
Sample size						0.680
<500	4		−0.056 (−0.291, 0.179)	0.001	82.1	
≥500	4		−0.066 (−0.133, 0.001)	0.043	63.2	
Children’s mean age						0.773
≤5 years old	6		−0.067 (−0.182, 0.048)	0.001	84.1	
> 5 years old	1		−0.010 (−0.230, 0.210)	/	/	
Child overweight/obesity						0.068
≤20%	3		−0.123 (−0.178, −0.068)	0.362	1.5	
>20%	2		−0.008 (−0.067, 0.052)	0.515	0.0	
Caregivers’ role						0.486
Mothers (only)	3		−0.095 (−0.244, −0.060)	<0.001	77.5	
Parents/grandparents/fathers	5		−0.022 (−0.145, 0.101)	0.012	82.5	
Caregivers’ education						0.237
Less than half with college degree or higher	2		0.038 (−0.236, 0.312)	<0.001	94.8	
More than half with college degree or higher	5		−0.102 (−0.184, −0.021)	0.169	37.8	
Family income						0.266
High-income percentage ^a^ below median level ^b^	1		0.010 (−0.070, 0.090)	/	/	
High-income percentage ^a^ above median level ^b^	2		−0.112 (−0.181, −0.043)	0.421	0.0	
**Concern about Child Overweight/Reward**						
Outcome measure		beta				0.750
CFQ	2		−0.063 (−0.119, -0.007)	0.758	0.0	
CFPQ	2		−0.027 (−0.214, 0.161)	0.454	0.0	
Country						0.750
Developed country	2		−0.027 (−0.214, 0.161)	0.454	0.0	
Developing country	2		−0.063 (−0.119, −0.007)	0.758	0.0	
Sample size						0.750
<500	2		−0.027 (−0.214, 0.161)	0.454	0.0	
≥500	2		−0.063 (−0.119, −0.007)	0.758	0.0	
Child overweight/obesity						0.787
≤20%	2		−0.071 (−0.138, −0.004)	0.587	0.0	
>20%	1		−0.050 (−0.150, 0.050)	/	/	
Caregivers’ role						0.638
Mothers (only)	2		−0.071 (−0.138, −0.004)	0.587	0.0	
Parents/grandparents/fathers	2		−0.040 (−0.129, 0.049)	0.661	0.0	
**Concern about Child Overweight/Restriction**						
Outcome measure		OR				0.413
CFQ	1		5.940 (1.740, 20.279)	/	/	
CFPQ	2		2.557 (2.012, 3.249)	0.817	0.0	
Exposure measure						0.951
Two/three items	1		2.460 (1.640, 3.690)	/	/	
One item	4		2.378 (1.502, 3.766)	0.041	63.6	
Country						0.289
Developed country	2		3.808 (1.781, 8.142)	0.366	0.0	
Developing country	3		2.141 (1.515, 3.027)	0.058	64.9	
Sample size						0.786
<500	2		2.337 (1.347, 4.055)	0.763	0.0	
≥500	3		2.520 (1.734, 3.663)	0.019	74.7	
Child overweight/obesity						0.656
≤20%	1		2.89 (1.098, 7.604)	/	/	
>20%	3		2.141 (1.515, 3.027)	0.058	64.9	
Caregivers’ role						0.951
Mothers (only)	4		2.378 (1.502, 3.766)	0.041	63.6	
Parents/grandparents/fathers	1		2.460 (1.640, 3.690)	/	/	

Notes. ^a^: the percentage of the highest income category in each article or the percentage noted with high-income/wealthy family; ^b^ the median percentage of high-income families from included articles.

## Data Availability

All data generated or analyzed during this study are included in the article and its Appendix A.

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
