# Peer review of "The Relationships between Caregivers’ Concern about Child Weight and Their Non-Responsive Feeding Practices: A Systematic Review and Meta-Analysis"

_nutrients, 2022, doi:10.3390/nu14142885_

Round 1

Reviewer 1 Report

Well-done on your research. This is a well-written manuscript on an important subject. Please see below my comments to improve the quality of your paper.

1.) Please include the databases searched in the abstract.

2.) What is the rationale behind your study? What will your study add to the body of knowledge? 

3.) You mentioned that your study aims to "...summarize the existing evidence on the associations between caregivers’ concern about child weight (including underweight and overweight) and their non-responsive feeding practices". What gap in the literature will this fill?

4.) If the inclusion criteria involved only children aged 1 to 11 years, than why include "Youth" (15-24 years) as a search term? 

5.) What are the study strengths?

6.) What is/are the study's policy implication(s)?

Author Response

We really appreciate your comments and guidance. Please see the attachment.

Reviewer 2 Report

It is a well-designed research, using correct systematic review and metaanalysis methodology. The paper reviews  the attitude of parents regarding the nutritional status in their kids. Although the topic has been reviewed recently regarding overnutrition, the authors consider both undernutrition and overnutrition. The exposition is clear and conclusions are solid and based on the results of their research

Author Response

We really appreciate your comments. Please see the attachment.
